# Community Based Interventions for Problematic Substance Use in Later Life: A Systematic Review of Evaluated Studies and Their Outcomes

**DOI:** 10.3390/ijerph17217994

**Published:** 2020-10-30

**Authors:** Trish Hafford-Letchfield, Tricia McQuarrie, Carmel Clancy, Betsy Thom, Briony Jain

**Affiliations:** 1School of Social Work and Social Policy, University of Strathclyde, Glasgow G4 OLT, UK; 2Department of Mental Health and Social Work, Middlesex University, London NWA 4BT, UK; t.mcquarrie@mdx.ac.uk (T.M.); c.clancy@mdx.ac.uk (C.C.); b.thom@mdx.ac.uk (B.T.); briony.jain@gmail.com (B.J.)

**Keywords:** ageing, older people, problematic substance use, interventions, evaluation, care services

## Abstract

Problematic substance use (PSU) in later life is a growing global problem of significant concern in tandem with a rapidly ageing global population. Prevention and interventions specifically designed for older people are not common, and those designed for mixed-age groups may fail to address the unique and sometimes complex needs of ageing communities. We report findings from a systematic review of the empirical evidence from studies which formally evaluated interventions used with older people and reported their outcomes. Nineteen studies were included, of which thirteen focused solely on alcohol-related problems. Eight interventions utilised different types of screening, brief advice and education. The remaining drew on behavioural, narrative and integrated or multi-disciplinary approaches, which aimed to meet older people’s needs holistically. Quality assessment of study design helped to review evaluation practice. Findings point to recommendations for sustainable and well-designed intervention strategies for PSU in later life, which purposefully align with other areas of health and well-being and are delivered in locations where older people normally seek, or receive, help. There is further scope for engagement with older people’s own perspectives on their needs and help-seeking behaviours. Economic evaluation of the outcome of interventions would also be useful to establish the value of investing in targeted services to this underserved population.

## 1. Introduction

Age-related aspects of problematic substance use are an important public health challenge due to the incremental number of people affected [1]. There is a lack of consensus in the literature on the terms used to describe ageing [2]. The World Health Organizations’ review of age classification observed wide variation between countries and over time. They suggested that transition to becoming “older” occurs between the ages of 45 and 55 years for women and between the ages of 55 and 75 years for men [3]. In this paper, we included studies with age cohorts starting from 45 years to capture the full age range when discussing problematic substance use (PSU) and ageing. We also used the broad term “problematic use” to capture dependent, recreational and/or prescribed use of drugs and/or alcohol, which negatively impacts on the user’s life either socially, financially, psychologically, physically or legally [4].

Currently around 962 million people worldwide (13%) are aged 60 years and over and this is projected to rise to 1.4 billion by 2030 and 3.1 billion by 2100, [5]. These trends highlight the urgency of addressing problematic substance use [3,6,7,8,9]. Other trends such as the impact of medical advances on extending our lifespan, the changing nature and use of different substances on the market are all influences on how problematic substance use influences and affects ageing, and the specific needs of older people for support services [4,10].

### 1.1. Problematic Substance Use in Later Life

Older people who use substances can be categorised into two distinct groups: early onset users (survivors) or late onset users (reactors) [7]. The early onset users have a previous and longer history of substance use that continues into later life. Late onset users on the other hand may begin to use substances later on in life perhaps following a stressful event such as bereavement, retirement or due to social isolation. Recommendations on how to treat this latter high-risk population appear to be least discussed in the literature. Treatment programmes tend to focus on the “survivors” who may already be within opioid substitution therapy (OST) programmes. The late onset users, however are a larger but less visible group, who are at risk of being neglected and of placing greater strain on services [6,11].

The demand on health and social care of people aged 65+ years needing support for problematic substance use is well documented [6,11,12]. For example, problematic use of alcohol and other drugs (AoD) including polypharmacy (over-the-counter and prescribed medication) is associated with increased use of emergency services and hospital admission [7,13,14,15]. Older people with co-occurring conditions have been shown to experience delayed transfers after hospital admissions, premature transfer to long-term care and present more frequently with adult abuse [16,17,18]. For those with refractory alcohol problems, there is likely to be a growing demand for long-term specialist care [19]. In most countries, these issues are being addressed amid tough financial conditions and a fragile care market, creating unprecedented pressures on primary care and secondary care services [20,21,22]. Given the projected increase in older populations, there may be a sound health-economic argument for greater investment in services to make best use of available resources.

### 1.2. What Works for Older People with PSU?

Whilst there is a growing body of evidence on the prevalence, type and impact of problematic substance use in later life, there is still no consensus on what works best or on evidence-based interventions [10]. Nicholas et al. [23] articulate how alcohol and other drugs (AOD) use among older people occurs along a spectrum. At one end are individuals who do not use any alcohol or drugs. Among those who do use alcohol or drugs, some people experience non-problematic use for example during recreational use. Others may develop a range of problems such as complications with excessive alcohol consumption combined with prescribed or over-the-counter mediation particularly where there are underlying health conditions and other co-morbidities. Individuals can move backwards and forwards along this spectrum of use. This requires a corresponding spectrum of support, ranging from preventative, low-threshold and person-centred early interventions to more comprehensive treatment programmes which respond to people with multiple morbidities [24].

There appear to be few targeted, tailored AOD services for older people and access to problematic substance use services can be difficult. This often means that other health and social care services may become the default treatment and support option. Older people with problematic substance use may present for treatment across different care pathways, primary care, mental health services, old age services, care homes etc. In conclusion, not having sufficient information about what works for older people with problematic substance use may lead to poorer outcomes, including poor access to interventions, higher rates of relapse, higher costs of care and poorer treatment engagement [25]. It would seem crucial therefore to review the relevant evidence to understand what interventions are used to tackle this widespread problem of older people and problematic substance use. Secondly, for those outside of specialist hospital-based treatment programmes, we were particularly interested in what happens in the community and particularly how those “community-based” services are best placed to respond to older people. Focusing on community-based interventions is valuable for describing the full range of interventions-prevention, early intervention and programmes of support outside of specialist inpatient hospital care.

### 1.3. Aims and Objectives

The primary aim was to identify and synthesize research on the outcomes of interventions for problematic substance use in later life provided in the community. Synthesising this body of evidence can help to identify gaps in knowledge and suggest recommendations for further research and intervention. The detailed aims were:To identify community-based interventions including preventative and early intervention programmes used with older people with problematic substance use.To collate evidence on the range and type of interventions used.To identify methods used to evaluate the programmes, interventions used and their effectiveness.To describe the findings on programme effectiveness and utilization in different care settings.

## 2. Materials and Methods

The protocol for the systematic review was preregistered with PROSPERO an international prospective register of systematic reviews and this protocol can be accessed online [26].

### 2.1. Search Strategy

The review was conducted in accordance with the Preferred Reporting Items for Systematic Reviews and Meta-Analyses (PRISMA) guidelines [27]. Electronic databases were searched using keywords and MeSH terms. References from relevant articles were scanned to identify other relevant sources. Table 1 documents the search strategy, data bases and search terms used.

### 2.2. Inclusion and Exclusion Criteria

Studies included were specific to people aged 45 or above with problematic substance use. We included peer reviewed studies with age cohorts starting from 45–60 years, to capture the full age range. Studies on ageing often use age stratification to compare between age cohorts and to contextualise issues within the environment and culture of different groups [28]. Table 2 provides full details of the inclusion and exclusion criteria.

### 2.3. Study Selection

Whilst there is a range of grey literature that may address interventions, we did not include these as we were not able to establish whether they had been peer reviewed. Studies were screened following the Participants, Interventions, Comparisons, Outcome, Study design (PICOS) eligibility criteria (available in Appendix A) [29]. Nine databases were searched and 2690 study titles and abstracts from the search results were screened by one researcher (author 2) removing those which did not meet the inclusion criteria (2444 abstracts). Studies meeting the inclusion criteria (246) were imported into the online systematic review management system software COVIDENCE (https://www.covidence.org/) for detailed screening by authors 1 and 2. COVIDENCE automatically removed 57 duplicates leaving 189 studies to be screened by two members of the research team. A further 96 studies were excluded after a second screen of titles and abstracts due to not meeting the inclusion/exclusion criteria. Selected studies (93) were subject to full-text review. Any doubts regarding inclusion/exclusion were discussed and resolved by the team. The main reasons for exclusion of studies at the 3rd screening stage were: primary focus was on prevalence of problematic substance use or patterns of use; tobacco smoking as main substance; absence of clearly stated interventions and evaluation strategies; a focus on professionals or no focus on community-based interventions. Some studies included participants who crossed the age boundary from a younger middle age to later life making it difficult to isolate the data for our target group and these were excluded (See Figure 1).

### 2.4. Data Extraction and Synthesis

Detailed data was extracted from 19 included studies according to the pre-agreed extraction criteria outlined in the PROPSERO protocol (Authors 1, 2 and 5). The data extracted focused on study characteristics such as study design, target population, participant numbers, participant characteristics, type of intervention, substances targeted, outcomes measured, key findings and recommendations. Key features of the interventions, type of evaluation, screening tools, the outcomes measured, and statistically significant results were also extracted.

### 2.5. Quality Assessment

Two quality assessment tools were used to assess included studies; The Medical Education Research Study Quality Instrument (MERSQI) and the Critical Appraisal Skills Programme (CASP) Randomised Controlled Trial Checklist [30,31]. The MERSQI has a clear validated assessment scoring system with items such as study design, sampling, type of data, validity of evaluation instrument, data analysis and outcomes. Most reviews using MERSQI concern educational interventions [32,33,34]. MERSQI was considered relevant for this review given the number of studies using education in the interventions evaluated. It also accommodated observational or experimental study designs. All items in each domain are scored on a scale of 1 to 3 and added up to determine a total MERSQI score, with a maximum score of 18.

Studies based on RCTs (*n* = 6) were further assessed using CASP simple critical appraisal checklist, for closer examination of the application of their findings for practice with older people [31]. The CASP checklist examines study design and covers three main areas: validity, results and clinical relevance. It enabled systematic consideration of three broad questions: Are the results of the study valid? What are the results? Will the results be locally relevant? Further details can be made available through the Appendix A.

## 3. Results

### 3.1. Overview of Study Characteristics

Table 3 provides a summative overview of the nineteen studies included in this review. The majority (14) were conducted in the USA, with the remaining five conducted in Canada (1), Denmark (1), UK (2) and Norway (1). All but one study targeted alcohol use, of which thirteen looked solely at alcohol and three targeted alcohol in combination with the use of over the counter (OTC) medications. The remaining three studies targeted AoD covering a range of substances including both prescription and illegal drugs. One study was concerned with polypharmacy relating to over-the-counter and prescription drugs. The study designs were mixed with RCTs comprising almost half (*n* = 8) which included two secondary analyses of data from existing RCTs. Two were cohort studies, one was a comparison study, six used pre- and post- intervention questionnaires and two were qualitative studies.

### 3.2. Quality Assessment

As previously outlined in Section 2.5, two quality assessment tools were used to assess included studies; the MERSQI assessment tool and CASP Randomised Controlled Trial Checklist [30,31].

#### 3.2.1. MERSQI Assessment

The range of scores for the MERSQI quality assessment was 7.5 to 18; the details of the individual study scores are reported in Appendix A. The main reasons for a lower score were absence of objective measures or not reporting on the validity of measures, and absence of measures to capture focused outcomes such as demonstrable change in the behaviour, health and wellbeing of study participants. Furthermore, two of the lower scoring studies reported greater emphasis on study participants’ descriptive or self-reported outcomes [37,40].

In relation to validity, not all of the studies used validated measures; for example, in an educational intervention, researchers designed their own pre- and post-test assessment measures based on a literature review and to reflect their study aims and objectives [37]. This served to assess participants’ level of knowledge gained and tested attitudes using a Likert scale to rate value statements. Most of these were measures over a short period. Also looking at the impact of education, one study [40] used a pre- and post-test, consisting of the same 16 knowledge items plus additional questions in the post-test; the test looked at what had surprised them or influenced them to do anything differently. Some of the outcomes measured were very relevant such as enhanced knowledge of problematic substance use, engagement with services and reduction in use [38].

Fourteen of the studies reported on the validity of the measurement instrument used. For example, a measure of health-related quality of life outcomes (HRQL), included measuring mental and physical health using the Geriatric Depression Scale (GDS) to assess how the patient was feeling [36]. They also used reverse coding to measure depressive symptoms when following up the patient after 2 months. Across the studies, the validated tools used to measure outcomes included the following:Michigan Alcoholism Screening Test-Geriatric Version (MAST-G)Global Assessment of Functioning (GAF)Health Screening Survey (HSS)Alcohol Use Disorder Identification Test (AUDIT)Mental Health Inventory (MHI)MOS SF-12, Basis 32, and Substance Abuse Inventory (SAI)Older American Research and Service Centre Instrument (OARS)

Within those studies that implemented more complex interventions, a range of measures were used, such as reduction in drinking, quality of life and resources used. Using as the primary outcome the average drinks per day (ADD) derived from an extended AUDIT-Consumption (3-item) (AUDIT-C) at 12 months—One study [53] identified secondary outcomes using the AUDIT-C score at 6 and 12 months; alcohol-related problems were assessed using the Drinking Problems Index (DPI) at 6 and 12 months. The researchers also assessed health-related quality of life using the Short Form Questionnaire-12 items (SF-12) at 6 and 12 months; ADD was assessed at 6 months; quality-adjusted life-years (QALYs) (for cost–utility analysis) was derived from European Quality of Life-5 Dimensions; and health and social care resource use.

Within the qualitative studies, the researchers drew on their analysis of field notes and transcripts, and were able to identify key themes regarding what participants found helpful about their group narrative therapy process [50]. Another study that was looking at the impact of providing a wet care home environment, noted that processes were not in place to collect quantitative measures in a way that could provide clear evidence of impact and that there was no single outcome instrument to capture outcomes for this type of provision, which was more holistic [46]. However, the researchers used a combination of standard tools administered on admission and at regular intervals thereafter to give a rounded picture across multiple domains. Whilst this study also relied on mostly qualitative measures, they also designed measures to assess the outcome of interventions on the individual and process outcomes. These focused on the impact of service delivery and other system factors, which were relevant to harm reduction for older drinkers with refractory problems. Using the theoretical underpinning of Appreciative Inquiry also helped to focus on the root causes of what works, why it works and how it works, rather than focusing on problems of individual older drinkers [30].

Studies that had the highest MERSQI scores were RCTs. In the context of problematic substance use, RCTs enabled comparison between populations where one group was allocated to receive an intervention and the other to receive a control. For example, one study compared two groups of older men who have sex with men, who had problem drinking and high risk of HIV transmission, with a general population sample with problem drinking [44].

#### 3.2.2. CASP Randomised Controlled Trial Checklist

Studies based on RCTs were assessed more closely, using the CASP simple critical appraisal checklist, in relation to the application of their findings for practice with older people. The scores for the three main areas: validity, results, and clinical relevance are reported in detail in the Appendix A.

Although the quality of the RCTs methodologies varied slightly, most of the included studies scored well against the items on the checklist suggesting that the trials were carried out to a good standard. One example was how focused the trial was, on its original aim. The assessment revealed that all of the trials had a very clearly focused aim. The RCT methodologies also scored highly in relation to the randomization to treatment arms and all accounted for the total sample. However, what was lacking across most of the trials was a precise estimate of the treatment effect with few reporting confidence intervals. The types of participants varied across the trials; some had more complex presentations and would not usually be seen in primary care or substance use settings [39] whereas others involved participants who were generally in good health [36] so results could have been affected by the severity and complexity of problematic substance use and co-morbidities. Only one of the studies blinded the participants and health workers [42]. As discussed already, there was also great variation in the way the treatments were delivered. All but one study reported whether there were statistically significant differences between groups or not [39].

### 3.3. Defining and Assessing Problematic Use

The definitions and tools used for assessing problematic use prior to the intervention varied, making comparison between studies difficult. Examples from the tools used in included studies were; Substance Abuse and Mental Health Services Administration (SAMHSA, 2004) [37,49]; National Institute of Alcohol Abuse and Alcoholism Guidelines (NIAAA, 2004) [38,52]; Michigan Alcoholism Screening Test-Geriatric version (MAST-G) [39,45,51]; Co-morbidity alcohol risk evaluation tool (CARET) [36]; Alcohol Use Disorder Identification Test (AUDIT) [41,47]; Alcohol related problems survey (ARPS) [41]; Alcohol Dependence Scale (ADS) [44]; Substance Abuse Inventory (SAI) [48]; Health Screening Survey [42].

Studies evaluating educational interventions for prevention, designed their own questionnaire or checklist to assess use [35,40,50]. The UK studies used ADD/AUDIT-C and MAST-G [51,53].

### 3.4. Populations Studied

Most studies centred on interventions addressing older peoples specific needs or were targeted at those perceived to be at risk [35,37,40]. These were underpinned by a belief that older people, if motivated, are capable of addressing their own needs or reducing their risk if provided with appropriate education and support [37]. Recruitment selection processes involved active outreach or going to places where older people already were, such as “senior community centres” or housing schemes [49,50,52] and primary care settings [41,42]. Benza et al. specifically accessed “housebound” older people and Outlaw et al. focused on outreach to African Americans in settings such as barber and beauty shops, churches, and a Black medical school. As a result, Outlaw et al.’s sample [49] included 34% African Americans, compared with the other studies which reported the lack of racial diversity and other cultural characteristics in their samples. Age and diagnostic category were common criteria for eligibility.

Specific groups targeted included “men who have sex with men” (MSM) and drank problematically (the latter behaviour considered to carry higher risk for these men of HIV transmission) [44]; older veterans [38,47]; and older women perceived to rarely or be less likely to access services [40]. Oslin et al. [47] examined differences in the clinical presentation and treatment outcomes of middle aged compared with older people in a prospective naturalistic study of patients admitted to a residential rehabilitation center for alcohol dependence. McCann et al.’s [46] intervention was with people whose needs could not be met adequately in mainstream care homes, many of whom had cognitive impairment including Korsakoff syndrome, issues with continence, limited mobility and challenging behaviour. Outside of this study, no other studies included people with a diagnosis of dementia or homelessness, due to the need to obtain longitudinal data and to increase the reliability of data.

Besides active outreach, some studies used secondary data. For example, convenience sampling via the Danish Civil Registration System was used in one study to administer an internet-based questionnaire; a health examination, secondary data and the Alcohol Use Disorder Identification Test (AUDIT) were used to identify participants with heavy drinking [47].

### 3.5. Overview of Interventions and Key Outcomes Measured

Table 4 provides an overview of the study settings, interventions and key outcomes measured in the evaluation.

#### 3.5.1. Interventions

Interventions fell broadly into the following categories; screening, identification and brief interventions, some of which combined education, motivational interviewing and counselling; educational interventions; therapeutic interventions in the form of group therapies (reminiscence and narrative); and individual therapies, which addressed cognitive-behavioural and bio-psychosocial factors. Other interventions used stepped-care approaches such as 12-step harm reduction programmes and more holistic approaches that targeted a wider range of needs in addition to PSU delivered by interdisciplinary and integrated teams. Provision of a specialist “wet” care home also addressed wider, complex needs of people requiring residential care. The intervention approaches have been mapped to Nicholas and Roche spectrum of problematic substance use [23] in Figure 2 to show the full range of responses corresponding with patterns of use in later life and to set the context for further discussion on the specific approaches used [54].

#### 3.5.2. Educational Interventions

Educational interventions included a group intervention called “prevention Bingo”, (which was based on a popular game often used in group environments) [37] and online courses [35,40,42]. Eliason and Skinstad described a one hour taught education programme on alcohol covering definitions, myths and attitudes, how alcohol and drugs interact with metabolism and ageing, with the aim of introducing alternative behaviours to avoid drinking and to promote well-being. Alemangno et al. [35] designed short video clips on medication misuse and their possible risks and provided a medication checklist for participants to share with their doctor. Some educational interventions were also combined with other interventions such as alcohol counselling and providing patients with a personalised report [36,41]. Education was also a component of the other interventions described below.

#### 3.5.3. Brief Interventions

Brief interventions (often known in shorthand as BIs) varied in content and delivery but with a key aim of encouraging further and future uptake of health services through providing assessment and direct feedback, goal setting and contracting [38,42]. Some interventions included education, with an information booklet and self-screening questionnaire and motivational interviewing [38,43].

Fleming et al. [42] conducted a community based RCT testing the efficacy of physicians giving advice on reducing alcohol use. Two × 10–15-min counselling sessions were delivered to an intervention group of 87 patients (control group = 71) including advice, education and behaviour contracting, using a scripted workbook. Rao’s study [51] described an intervention provided by community nurses offering brief advice which involved a conversation lasting 2–5 min, during which there was an estimation of drinking patterns, awareness of risks, benefits of cutting down or stopping and advice on how to achieve goals. Participants identified as positive for high-risk drinking behaviour were subsequently invited to engage in a more detailed assessment that addressed factors contributing to increased alcohol use. Community nurses were instrumental in delivering structured feedback, facilitating plans and assessing motivation to change drinking behaviour, with progress reinforced and reviewed to enhance reduction in high-risk drinking. Watson et al. compared the clinical effectiveness and cost-effectiveness of a stepped care intervention versus a Brief Intervention in the treatment of older hazardous alcohol users in primary care. The minimal intervention group received a 5-min brief advice intervention with the practice or research nurse delivering feedback of the screening results and discussion regarding the health consequences of continued hazardous alcohol consumption. Those in the stepped care arm initially received a 20-min session of behavioural change counselling, with referral to step 2 (motivational enhancement therapy) and step 3 (local specialist alcohol services) if indicated [53].

Schonfeld et al. described seven different ways of enhancing Brief Interventions (to include depression and suicide risk, as well as prescription, OTC medication and alcohol use), by asking provider agencies to carry out screening in different types of services including health care, ageing, senior housing, mental health, and substance misuse treatment services. Participants could receive several BI sessions based on individual need [52]. Another study trialed the implementation of a Brief Motivational Intervention of short duration (average 11 min) in a general population-based sample of heavy drinkers and suggested that this could be a realistic approach for use in both primary health care and other settings [43].

#### 3.5.4. Treatment Approaches Provided in Community-Based Facilities

Other types of interventions were delivered in community-based facilities (see Table 4, column 3 for the nature of settings). They sought to adapt treatments for the needs of the older population, to maximise flexibility and address related issues. One intervention followed a cognitive-behavioural and self-management treatment approach specified by a manualized curriculum. This drew on relapse-prevention models that aimed to teach older people how to identify and cope with high-risk substance use situations. The curriculum had nine modules covering management of internal and external stressors associated with PSU, and helping the older person to recognise and manage negative feelings to help build coping strategies. This treatment was adapted in a variety of settings, both with individuals and with groups supplemented by individual therapy sessions, case management services, and medication management by a staff psychiatrist and a nurse practitioner. Therapists adapted the curriculum for people who wished to participate in the program individually but not in a treatment group [49]. Another manualised intervention used a problem-based approach by addressing older people’s presenting concerns through psychotherapy or psychopharmacology. This had a strong educational component for both service users and care coordinators. Psych-education included discussion of diet, exercise, the health effects of alcohol and tobacco use, and techniques for making healthier lifestyle choices. Care coordinators were encouraged to eliminate as many barriers to care as possible including simplifying appointment schedules, arranging transportation, assisting with financial or legal concerns, and helping to connect people with senior services in the community. Close follow-up by phone or in person supported treatment at whatever intensity or duration was required [47]. Those enrolled in Oslin et al.’s UPBEAT programme [48] also received a comprehensive psychogeriatric assessment administered by a multidisciplinary team, which included geropsychiatry, geropsychology, social work, and/or nursing. Coordination of care involved conducting a thorough clinical assessment, patient engagement and assisting the patient in adhering to the treatment plan. The treatment centre had disability access, and the programme conducted at a slower pace than “treatment as usual” with age appropriate group work. Another community-based programme, designed to meet the needs of older people experiencing significant crises related to substance use, mental and physical health problems, provided interventions that addressed access barriers for those who were isolated or “in home” contexts and unable to reach services [39].

Further Poole et al. [50] documented how therapists drew on the tenets of narrative therapy to assist participants to develop, strengthen and communicate their identities in relation to problematic substance use and mental health issues in a group setting. Eight weekly sessions with twelve older people, using a strengths-based and life course approach, in which individuals were able to tell their stories, facilitated the valuing of individual accomplishments and gave them a set of tools to combat their problems.

#### 3.5.5. Holistic Interventions through Service Design and Delivery

A number of studies addressed themes on multi-morbidities, in particular mental health and general well-being of older people with problematic substance use, and described an approach, which embedded holistic approaches within the interventions examined. Rao’s London based study [51] addressed the dual issues of problematic alcohol use accompanying other mental health disorders. A multidisciplinary community mental health team in a challenging socio-economic environment provided integrated care where there was a high prevalence of problematic alcohol use (PAU) and mental health. The service had four community psychiatric nurses, one of whom had specific expertise in problematic alcohol use and worked alongside a consultant old-age psychiatrist in the assessment, treatment and provision of aftercare for older people with co-occurring mental health and substance use problems. A standardised generic assessment tool was supplemented by questions covering alcohol use, particularly estimation of quantity/frequency of alcohol intake; this was combined with a more specialised assessment covering the person’s history of substance use and any associated physical and mental health problems, which in turn allowed identification of factors precipitating and maintaining alcohol use. Assessment led to diagnosis and a treatment plan. The strengths of this approach are that it enabled the tailoring of services to the older person’s occupation, relationships, socio-economic issues and history of mental health. Home visits by the community nurse enabled monitoring of alcohol use, assessment of level of function, monitoring of nutrition and compliance with medication and linking the person into their local community. The team also had a support and recovery worker working alongside the specialist nurse to accompany patients to local amenities such as day centres and primary care surgeries and to facilitate use of advocacy services to meet social and welfare rights.

The holistic approach described in Rao’s study [51] involved family and carers using the Care Programme Approach (CPA) common to mental health services in the UK providing expert assessments in mental capacity, adult safeguarding and expertise in mental health. Screening for the presence of concurrent mental health difficulties and substance use, together with developing and sustaining collaborative therapeutic relationships with patients and constructing care plans designed to address these needs were found to be successful in implementing simple, low-intensity evidence-based interventions safely and effectively in partnership with patients. The integrated care approach helped to identify patients whose needs are sufficiently complex to require high-intensity interventions.

Another example was McCann et al.’s [46] study of people in two wet care homes in UK and Norway which uniquely address the needs of older people who may be difficult to place and unable to commit to the expectations and demands of mainstream care homes. Wet care homes focus on harm reduction strategies to reduce harm from high-risk alcohol use, rather than insisting on abstinence. Harm reduction entailed full on-site personal care services designed specifically for people with refractory alcohol problems.

These two interventions were unique in how they took account of physical, emotional, psychological and wider factors and the interrelationships between older people’s health, well-being and use of substances.

### 3.6. Key Outcomes Measured

Outcomes from the interventions are summarised in Table 4. A range of recovery focused outcomes were recorded including abstinence from substances, reduction in the use of substances, harm minimisation, combined improved physical, psychological wellbeing and enhanced knowledge of problematic substance use and its management. The studies focused on what was learned from intervening with problematic substance use in later life, including: the value of education, from raising awareness for prevention to facilitating greater engagement with managing their situations; the relevance of working with problematic substance use alongside other factors that impact on ageing in relation to implementing Brief Interventions; providing more holistic treatment approaches and thinking about the best way to tailor or target problematic substance use services to older people by adapting their design and delivery and including cost-effectiveness of intervention.

#### 3.6.1. Education and Brief Interventions

In relation to Brief Interventions and educational interventions, like other age groups with problematic substance use, older people generally responded well to the intervention, for example they increased awareness or corrected myths about risks of alcohol on use of over the counter and prescription drugs [40]. This, however, did not always lead to additional help seeking beyond the intervention possibly because the intervention itself was sufficient, or because of continuing social stigma regarding using mental health care, or even because people preferred to rely on their own resources [38,40]. Further, Brief Interventions might need to be tailored differently for people who are primarily ‘binge’ drinkers rather than steady drinkers.

When following up trends in alcohol consumption following Brief Intervention, Gottlie-Hansen et al. [43] found no consistency in alcohol reduction. There was speculation on whether people with serious existing problems might be less motivated or less likely to attend an educational programme. Fink et al. [41] found that older primary care patients can effectively reduce alcohol consumption and alcohol use patterns when given personalised information reports about their drinking and health. Providing analogous information to physicians, as was done in their combined intervention, is effective in decreasing total alcohol consumption, but is no more effective at decreasing the associated risk (as measured by drinking classification) than personalised information reports only to patients. Physicians typically focus on achieving decreases in quantity and frequency of use rather than on alcohol’s interaction with overall health, medication, and functional status. In older adults, the amount of dysfunctional alcohol use causing physical, psychological, and social harm may be as important as the amount and frequency of alcohol consumption. Fink et al. found that the main outcome was change in drinking classification at follow-up. Improvement occurred when patients eliminated drinking risks; for example, their classification changed from harmful to hazardous or non-hazardous drinking. Patients accomplished this change by altering the balance between their alcohol use and their health, medications, behaviour, or functional status. Barnes et al. noted that physicians were initially concerned about incorporating a personalised patient report on older people’s drinking risks due to their own time challenges. However, physicians reported that they found it valuable and that it did not constrain their ability to discuss other medical conditions during the patient’s visit [36].

Finally, it was difficult to establish the efficaciousness or effectiveness of Brief Interventions, perhaps because follow up was often short term (3–6–12 months) [44]. Han & Moore identified that Brief Interventions with older people require attention to the language used and the unique physiological and social changes that occur in ageing [55]. Gottlieb Hansen et al. [43] found that Brief Motivational Intervention (BMI), even with booster sessions to maintain behaviour change efforts, had no effect in reducing alcohol consumption and the quality of its delivery was sub-optimal. Kuerbis et al. looked at 3 RCTs for Brief Interventions and results for the timeline follow back interview, which assesses quantity and frequency of alcohol pre and post treatment. They compared basic demographics and severity of alcohol use and negative consequences of four groups of middle age and older patients who received and did not receive treatments. Whilst not conclusive, they found that moderation was not possible for older people with hazardous problematic drinking and that there is a role for stepped care interventions or alternative skills-based interventions with this population [44].

Some studies compared outcomes for different groups. A study which compared Brief Intervention with a stepped approach found that both groups reduced alcohol consumption between baseline and 12 months, although the difference was not significant [53]. Schonfeld et al. [52] compared substance use and SBIRT (Screening, Brief Intervention, and Referral to Treatment) services for older adults screened by the Florida BRITE (Brief Intervention and Treatment of Elders) Project across 4 categories of service providers in non-health contexts (problematic substance use treatment, behavioural health and aging services). They found that health educators screening solely within medical sites recorded fewer positive screens than those from mental health, substance use, or aging services that screened in a variety of community-based and health care sites. They also found that non-health care providers were more likely to follow up participants and there was improved targeting or referrals of high-risk older people identified by community programs, compared to those identified in universal screening of health care patients. Mental health and substance use agencies recorded greater percentages of non-treated individuals following a positive screen than did aging and healthcare agencies. Healthcare agencies tended to report on patient depression more frequently, perhaps related to patients with serious, concomitant medical problems. Six-month follow-ups revealed significant reductions in substance use. Results suggest that SBIRT is a low-cost, effective strategy to address older adults’ risky use of substances, especially when combined with outreach and screening methods used where elders reside or receive various services. For instance, Rao’s study [51] of integrated care provided by a multidisciplinary community mental health team, confirmed that 38% of the 50 patients seen achieved abstinence from alcohol or controlled drinking at the 6-month follow-up stage.

Oslin et al. found few differences in outcomes between middle age and older people attending a residential rehabilitation community programme for alcohol dependence. Whilst older people did significantly better in their treatment response, they were less likely to engage in formal aftercare and this was interpreted as a predictor for poorer longer-term improvement. This was another example where older people perceived their addiction as not severe enough to warrant aftercare; but also there may be barriers to follow up such as access, transportation and financing. Further, the use of alternatives such as telephone, internet or interactive voice recordings were not as easily utilised by people in later life [47].

Oslin et al., UPBEAT Program, a clinical demonstration project conducted across nine Department of Veterans Affairs Medical Centers (VAMCs), aimed to demonstrate the cost-effectiveness of case identification and treatment of older veterans suffering from depression, anxiety, and/or problematic alcohol use who were not currently engaged in formal treatment. As such, the focus of the project was on the long-term management of elders. Preliminary findings found that patients randomized to UPBEAT care in a previous study had significantly less costs associated with hospitalizations than did those randomized to usual care. However, overall costs including outpatient costs were similar between the 2 groups. Oslin et al. specifically examined the impact of UPBEAT (compared to usual care) on symptom reduction and quality of life at the patient level and found very little different in uptake and outcomes [48]. Watson et al. [53] calculated the overall average cost per patient. Taking into account health and social care, there was very little difference in resource use when comparing the stepped care group and minimal intervention group at month 6. The mean QALY gains were slightly greater in the stepped care group than in the minimal intervention group. At month 12, participants in the stepped care group incurred fewer costs, than the control group. Therefore, from an economic perspective the minimal intervention was dominated by stepped care; however, as would be expected given the effectiveness results, the difference was small and not statistically significant. Whilst not focused on cost outcomes, McCann et al. [46] also suggest that although specialist placement in wet care homes is expensive, the costs may be offset by a reduction in the use of other health, social and criminal justice services.

#### 3.6.2. Older People’s Experiences

How the included studies addressed or captured older people’s own subjective experiences will be important to consider in relation to growing impetus in policy and practice regarding engagement with those using services in finding solutions. Guilt and shame were themes that many of the studies referred to in relation to reasons why older people did not take up opportunities for treatment, health care or other psychological support [50,52]. These included issues about labelling—both from those offering screening, advice and support and from those on the receiving end. Poole et al. illustrated that outcomes from narrative therapy, when a life has been led under the label of alcoholic or drug addict, enabled participants to externalize their problem and tell alternate stories. Outcomes from therapy were reported as empowering and supported by the building of friendship, peer support and the expression of other identities in later life.

#### 3.6.3. Studies Addressing Drugs

Most of the included studies addressed alcohol. In relation to those studies which addressed prescription drugs, Outlaw et al. [49] (measured after 6 months), found that people who completed the programme (attended 14 or more sessions out of 18) showed a significantly greater reduction in days of nonmedical prescription drug use and cognitive improvement in memory and concentration. They also experienced increased vitality, less pain and increased mental health. This reported outcome supports the literature that age-specific treatment programmes as well as programmes that are accessible to older people, help to maintain better health. Completers also had significant decreases in days of any alcohol use and days of binge drinking (five or more drinks in one sitting) within the past 30 days. The findings of this study demonstrated that actively taking treatment to where older adults live and socialize, such as their homes, religious institutions, and senior citizen centres, might improve their willingness to engage in aftercare. Overall, participants at follow-up reported significantly less stress in their lives, fewer emotional problems such as serious depression and anxiety, a decrease in having to reduce or give up important activities, and prescription of medication for psychological and emotional problems. This need to give attention to health would seem to have greater importance for older people. Oslin et al. [48] also identified somatic health as one particular challenge for older people with problematic alcohol dependence.

McCann et al. [46] findings from the residential care facility also found that the provision of regular nutritious meals, consistency in support, daily routines and providing a relatively safe environment with peer support and social activities contributed to improved quality of life including at the end of life. Negative outcomes reported were due to containment, boredom and residents reacting aggressively and causing conflict. These could result in adverse experiences for staff and other residents including verbal, racial and physical abuse making for a stressful environment.

## 4. Discussion

This review brought together a range of studies of interventions outside of hospital treatment programmes. These studies used formal measures to evaluate outcomes and provided insights into both the current state of research findings as well as highlighting gaps in knowledge about how to improve responses to the ageing population with problematic substance use.

Review of educational interventions demonstrated their value and helpfulness from a public health perspective. In addition, the provision of routine screening and provision of information about substance use may complement each other and, potentially, may be useful for harm prevention. Using a motivational interviewing approach to explore the advantages and disadvantages of reducing substance use, may enhance readiness for change; facilitating structured problem-solving may help to identify personal strategies to facilitate and sustain a reduction in substance use as well as encourage reflection on the need for change. Evidence from the use of personalised reports on drinking risks and how these interact with health as part of educational interventions for patients suggest that where primary care physicians have less time to spend, such reports can help older patients to persevere in monitoring and modifying their own consumption and other alcohol-related risks and problems [17,41]. Brief Interventions could be embedded more systematically within ageing support but, as seen in many of the studies, the complexity of health systems can make them difficult to implement [42]. The lack of training of the workforce, within health and social care, and to deliver screening or support, is also significant [56,57,58]. An important challenge emerging from the research is how to improve the effectiveness of brief interventions [43]. A meta-analysis of brief interventions found that the average duration of a brief intervention was more than 20 min and the research indicates that longer and shorter interventions achieve similar outcomes [59]. Wutzke et al. found that 5 min of simple advice was as effective as (a); 60 min of advice and counselling and (b); very brief (maximum duration of 15 min) single-session personalized-feedback interventions without therapeutic guidance [60,61]. The use of brief interventions also needs to focus on tension alleviation and the development of more positive coping strategies to enhance treatment models.

A strong theme was the importance of establishing trust and addressing the individuals’ primary care needs, taking a holistic approach, which does not separate out substance use from mental health issues, viewing them clinically as co-occurring disorders and attending to the legal and ethical aspects of care [51]. Many researchers, reflecting on their intervention study and its outcomes, highlighted the need for developing explicit protocols and guidance between services and professionals working with this population to ensure effective communication and information sharing across and between services, carers and family. They also noted a need for more specialist and non-specialist training, and for providing supervision and support systems for staff working in older people care where they may not be confident or aware of the issues and how to respond and refer on.

Some studies provided incentives for those administering interventions. These enabled structured evaluation; but they do not enable sustainability or mainstreaming within regular services in contact with older people. Whilst hospital care is essential for people with high risk, follow up and integrating ongoing support with a range of providers is important for continuing care [47,48,51]. This includes primary care and addressing barriers that limit access to a range of care, including mental and physical health. Looking at co-morbidities such as mental health and disabilities is important as older people with a lifetime history of problematic substance use are three times more likely to have a co-occurring mental health disorder, usually a depressive disorder [4]. Other research also suggests that older people, who have experienced an undiagnosed or untreated depressive illness, are at high risk for developing late-onset [62]. Again, some studies provided evidence of overcoming these barriers through the implementation of models of integrated medical and psycho-social interventions using a collaborative model.

The gap in reaching diverse communities remains. Active collaboration with older people and the use of life experience are coming to the fore in problematic substance use services. We know that culture, ethnicity, and gender influence problematic substance use and programs that include effective assessment, outreach, and intervention need to be designed, implemented and further evaluated to meet the needs of these target populations [63,64]. Methods such as narrative therapy which suit working with minority groups such as indigenous populations through traditional storytelling may be one means of achieving greater outreach [50]. There is a need for effective cultural and linguistically competent specialty programs targeting older adults with problematic substance use.

Finally, few studies provide any explicit theoretical basis for how interventions might respond to problematic substance use in later life. Two studies on theoretical frameworks to explore stigmatisation and strengths based approaches to change, perhaps highlighting a need to develop further theories or interventions [38,46].

## 5. Limitations

There were limitations in some of the research in relation to the small sample sizes, with opportunistic sampling in some studies, the complexity of the sample population in others and the lack of diversity in samples overall, which excluded people from the oldest old, or those who find it difficult to leave their home or who are reluctant to seek help in the first place. In common with research on ageing populations, some studies found follow up difficult due to reasons such as death, failure to complete all assessments, withdrawal of consent, being unable to contact participants or people moving away [47]. The quality of the data collected was also impacted where this was collected by operational staff who were not always trained for the purposes of the study (as opposed to researchers). Active outreach to older people can help overcome barriers to participation in both treatment and evaluation of treatment outcomes. It may also take longer to engage them and therefore assessment of their engagement may not be accurate within the limitations of a onetime research study. As in any community research, some researchers noted that participants tend to be healthier because severely ill and disabled persons are less likely to volunteer for enrolment in a study.

The quality of the design of some of the studies made it difficult to validate any observed effect. Those studies with stronger designs (randomized control groups, matched controls, or those incorporating additional design elements for eliminating threats to validity) were helpful in this regard. Secondly, there is an issue regarding use of instrumentation. Some studies, for instance studies that also looked at MH, relied exclusively on self-report measures without, for example, corroboration by others—for example family, other biological measures and other professional assessments. Those that used validated instruments or items from validated instruments to measure factors and their correlation to other factors, such as co-occurring disorders, found it very challenging to accurately and reliably distinguish between “symptoms” attributable to substance use and symptoms not attributable to substance use.

This review did not capture studies in progress or studies conducted outside of the peer review in the grey literature where there may be a lot of innovation in relation to intervention that reflects research in the real world rather than only published work [10].

## 6. Conclusions

There is a need for evidence-based programs designed for older people with problematic substance use, as well as improved access to mainstream treatment programmes. Interventions also need to address the full spectrum of problematic substance use including prevention. With the baby-boomer generation ageing, a term used to describe a person who was born between 1946 and 1964 and who are beginning to make up a substantial portion of the world’s population, especially in developed nations, there will be a significant increase in the numbers of older people with problematic substance use needing treatment by the next decade. This need may be understood in the light of expectations and generational experiences of a group of people who have a more liberal, lived experience involving the use of alcohol and other drugs, which may challenge social norms of what ageing is supposed to be about. Ethnicity, gender and culture must be considered when designing treatment programmes for older people as minorities and older people may be more concerned with stigma related to mental health and substance use treatment than other groups and thus not seek services for their substance use disorders [65].

It is difficult to generalize beyond local interventions and to bring forth important findings that support and inform practice and pose questions for further study. Many studies have described relatively short-term interventions and there remains a lack of knowledge regarding long-term management. There was also evidence of decreases in the use of substances in some instances, although it has been difficult to demonstrate sustainability. More coherent sustainable funding of research may be a factor.

What was clear was that most studies have suggested that specialised interventions for older people must address improvements in their mental health and social functioning to improve overall quality of life. Further work on the role of families and other social supports of older adults with problematic substance use and the role that they can play in supporting treatment, perhaps through education, was indicated, as well as more research on which professions are best positioned to deliver interventions such as Brief Interventions and education within more integrated models of support. Future research could address intentional versus unintentional use and/or problem use in relation to medication to help providers identify appropriate interventions [52] and explore the utility of community based care for patients with more severe symptoms who might show greater improvements after more intensive treatment [47].

Measuring outcomes is important to evidence impact and change for individuals with problematic substance use in later life, and to determine whether a particular intervention or service is working well and constitutes an effective response to a complex issue. A combination of standardised tools administered on admission and at regular intervals thereafter is needed to give a rounded picture across multiple domains as in the Outcomes Star study [51]. Finally, economic evaluation may be indicated for identifying potential cost savings and efficiencies for long-term outcomes as a result of interventions in later life. Measures that could provide clear evidence of impact or which allow full economic analysis could demonstrate to what extent these cost savings can be made, particularly in those holistic interdisciplinary services which have sought to reduce dependence on other services [12].

## Figures and Tables

**Figure 1 ijerph-17-07994-f001:**
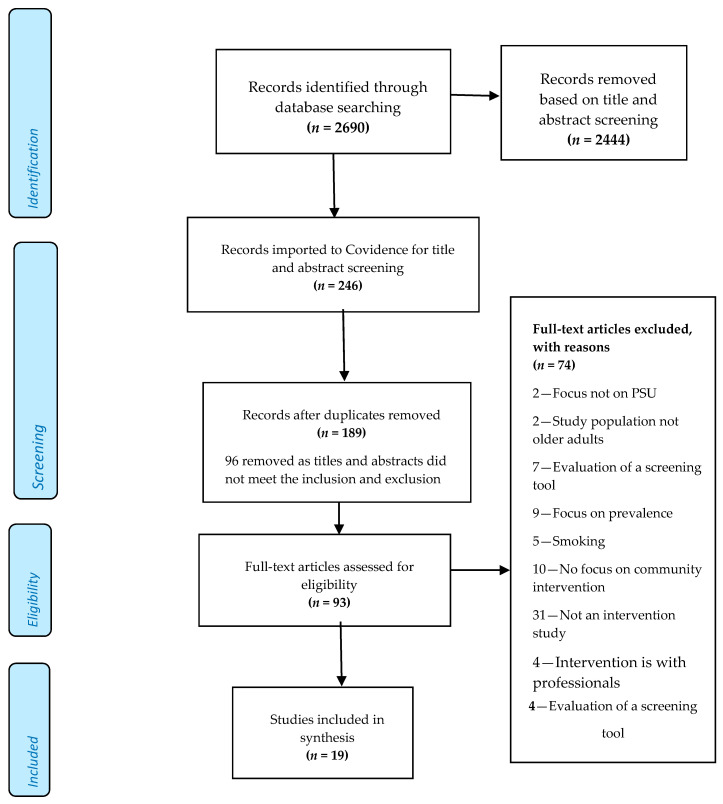
PRISMA flow diagram.

**Figure 2 ijerph-17-07994-f002:**
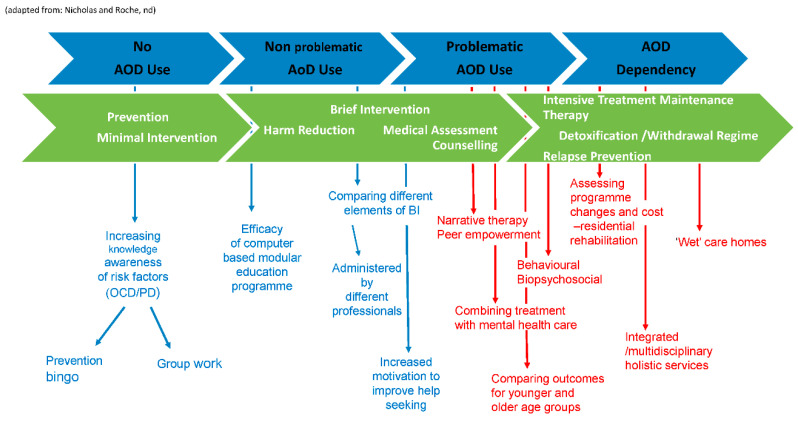
Mapping studies on the spectrum of problematic substance use.

**Table 1 ijerph-17-07994-t001:** Search strategy/databases, key words and MeSH terms used.

Databases searched	BioMed Central; CINHAL; Emerald; International Bibliography of the Social Sciences (IBSS); NICE evidence; OVID full text; PsycINFO; PubMed; Web of Science; MEDLINE; COCHRANE; British Nursing and social care online.
Keyword search terms	“old* people” or “old* adults” or elderly or ageing or aging or geriat* or geron* or mature AND addict* or “problematic substance use” or “substance misuse” or “alcohol misuse” or “alcoholism” or “drug misuse” or “drug abuse” or “alcohol abuse” or AOD or alcohol or “other drugs” or polypharmacy or “prescription drugs” or “non-prescription drugs” or narcotics or addiction or “dual diagnosis” or “drug depend*” or “alcohol depend*”
MeSH terms	“aged, 80 and over” or “aged” AND“substance dependence” or “substance addiction consequences” or “alcoholism” or “street drugs”

**NOTE:** MeSH terms may have varied in each database see Appendix A for full search strategy.

**Table 2 ijerph-17-07994-t002:** Inclusion and exclusion table.

Inclusion	Exclusion
Focuses on programmes for problematic substance use	Does not have problematic substance use as its key focus
Describes older people as the target population specifically or in comparison to the majority population	Target population is people under 45 years
The intervention delivered within community-based provision.	Is focused on ‘in-patient’ hospital only treatment
Has a clear description of the interventions used	Is not peer reviewed research
Has a clear empirical evaluation of the intervention/s	Does not contain evaluation of the intervention described
Qualitative, quantitative, review or mixed methods papers	Discussion documents
Describes outcomes of the intervention in its findings	Where the focus was on tobacco use only
Published in English	Published in a language other than English
Published between 1990 and 2019	Published before 1990

**Table 3 ijerph-17-07994-t003:** Overview of key study characteristics.

Source	Study Design	Country	Participants Age Range	Aims of Study	Substances Targeted
Alemagno et al. (2004) [35]	PP	USA	59–97	Test efficacy of educational computer programme to reduce medication misuse.	Prescription medication & OTC drugs
Barnes et al. (2016) [36]	RCT	USA	60+	To examine changes in health-related quality of life. Project SHARE interventions vs. TAU	Alcohol
Benza et al. (2010) [37]	PP	USA	60+	To develop and evaluate an educational programme to increase older adults’ knowledge of PSU.	Alcohol and OTC drugs
Copeland, Blow, Barry (2003) [38]	CS	USA	55+	Effect of BI on services use for older veterans who were at-risk drinkers.	Alcohol
D’Agostino et al. (2006) [39]	RCT	USA	51–91	To evaluate the Geriatric Addictions Program (GAP), designed to assist OA with PSU and DD.	Alcohol and OTC drugs
Eliason, Skinstad. (2001) [40]	PP	USA	54–91	Prevalence of AoD interactions in older women and if a BI would change knowledge.	Alcohol and OTC drugs
Fink et al. (2005) [41]	PCS	USA	65+	To evaluate whether providing physicians and older patients in primary care with personalized reports of drinking risks and benefits and patient education reduces alcohol related risks and problems.	Alcohol
Fleming et al. (1999) [42]	RCT	USA	65–75	To test the efficacy of BI in reducing alcohol use in older problem drinkers.	Alcohol
Gottlieb Hansen et al. (2012) [43]	RCT	Denmark	48–65	To test if a BI in a non-treatment seeking population of heavy drinkers results in reduced alcohol intake.	Alcohol
Kuerbis et al. (2013) [44]	Secondary analysis of data from 3 RCTs	USA	54+	Secondary data analysis of cases in three RCT’s that recruited problem drinkers, examining the effectiveness of BI. Additional comparisons to different age cohorts were made.	Alcohol
Lee et al. (2009) [45]	Secondary analysis of RCT	USA	65+	To assess the efficacy of a harm-reduction based intervention to enhance access to treatment and clinical outcomes among elderly at-risk drinkers.	Alcohol
McCann, Wadd & Gill Crofts. (2017) [46]	QS	UK & Norway	46–77	To describe the harm reduction models developed in two wet care homes in England and one in Norway.	Alcohol
Oslin et al. (2004) [47]	RCT	USA	60+	To examine the impact of the Unified Psychogeriatric Biopsychosocial Evaluation and Treatment (UPBEAT) Program, for elderly veterans	Alcohol
Oslin et al. (2005) [48]	PP	USA	50+	To examine differences in the clinical presentation and treatment outcomes of older adults with a diagnosis of alcohol dependence compared to middle-aged adults.	Alcohol
Outlaw et al. (2012) [49]	PP	USA	50+	To determine the effectiveness of the cognitive-behavioral and self-management treatment approaches targeted to older adults.	Alcohol, Prescription medication & OTC drugs & ID
Poole et al. (2009) [50]	QS	Canada	55–70	To review the effect of narrative therapy on OA coping with mental health and PSU.	Alcohol
Rao. (2014) [51]	CS	UK	65–85	To examine the outcomes of an integrated community nursing team for older adults with alcohol misuse.	Alcohol
Schonfeld et al. (2015) [52]	PP	USA	Mean age 66.5	Rolling the Florida Brief intervention and treatment for elders (BRITE project) out across 75 different sites.	Alcohol and ID
Watson et al. (2013) [53]	RCT	UK	55+	To compare the clinical effectiveness and cost-effectiveness of a stepped care intervention against a minimal intervention in primary care.	Alcohol

Abbreviations: OA, older adults; BI, brief interventions; PSU, problematic substance use; PP, pre-/post design; PCS, Prospective comparison study QE, quasi-experimental design; RCT, randomised control trial; QS, qualitative study; CS, cohort study; TAU, treatment as usual; P & OTC drugs, prescription and over the counter drugs; ID, illegal drugs; DD, dual diagnosis.

**Table 4 ijerph-17-07994-t004:** Study interventions and outcomes measured.

Source	Intervention Description	Setting	Participant	#	Outcome Measured	Results
1. Alemagno et al. (2004)	Education	Nine community senior centres	Majority female and white	412	Enhanced knowledge of PSU.	OA were more likely to use a medication reminder checklist and one third visited their doctor to discuss their medication misuse. No significance difference mentioned
2. Barnes et al. (2016)	Education	Primary care clinic	Majority white male	1049	Health and health related quality of life (HRQL)	A statistically significant effect on health and HRQL in the intervention group. Effects were most prominent for patients who received physician discussions.
3. Benza et al. (2010)	Education	Nursing homes and senior centres	Majority female	348	Enhanced knowledge of PSU.	A significant increase in knowledge regarding the risks related to medication and alcohol use.
4. Copeland, Blow, Barry. (2003)	BI	Primary care private sector and VA clinic	Majority white	205	Engagement with services	Significantly more veterans accessed medical outpatient services than those in the control group.
5. D’Agostino et al. (2006)	Targeted service for dual diagnosis	Community network/referral system	41 men (41.4%) 58 women	120	Treatment completion rates	The multidimensional motivational approach were more likely to result in treatment completion than the traditional referral approach. No significance difference mentioned
6. Eliason, Skinstad. (2001)	Education	Community senior day centre	All white Majority women	26	Enhanced knowledge of PSU	Participants’ knowledge increased post-test. The difference was statistically different.
7. Fink et al. (2005)	Education	Community primary care	All female Majority white	711	Alcohol consumption	Patients in the intervention group significantly decreased their alcohol consumption.
8. Fleming et al. (1999)	BI	Community based primary care practices	Majority male	158	Alcohol consumption. Number of binge drinking episodes. Health status.	Participants who received the BI demonstrated a significant reduction in 7-day alcohol use, episodes of binge drinking, and frequency of excessive drinking.
9. Gottlieb-Hansen et al. (2012)	BI	Community alcohol service	Mix of men and women	772	Alcohol consumption	There was no statistically significant effect of BI reducing alcohol consumption.
10. Kuerbis et al. (2013)	BI	Secondary data analysis of three RCTs	Male White	38	Alcohol consumption	OA responded to most interventions. Those who received brief evidence supported treatments were variable but mostly responsive. OA responded more strongly than MA with the exception of MI.
11. Lee et al. (2009)	Harm reduction vs. 12 step model	Community based alcohol service	Male 58% 50% non-Hispanic white; 35% African American	34	Engagement with services, alcohol consumption	Participants in the harm reduction arm showed a significant decrease in the number of drinks and number of binge drinking episodes. No significant changes in these outcomes in the 12-step model. Participants more likely to access treatment in the harm reduction group.
12. McCann, Wadd & Gill Crofts. (2017)	Provision of wet care home	Residential care homes	Mix of men and women	54	Impact of harm reduction, what works and why it works on wellbeing of residents	Themes included; Safety and security offered from risky and chaotic lifestyles. Regular health checks, reduced use of emergency services, lower risk of falls, Reduced alcohol use with some residents becoming abstinent and others moved on to detox and community alcohol treatment.
13. Oslin et al. (2004)	Unified Psychogeriatric Biopsychosocial Evaluation and Treatment	Department of Veterans Affairs Medical Centres	Majority white and male	2637	Behavioural health symptoms of older veterans.	No differences between UPBEAT and usual care patients on symptom or functional outcomes at any follow-up point Exploratory analyses suggested greater improvements in depressive symptoms in those assigned to UPBEAT care.
14. Oslin et al. (2005)	Targeted rehabilitation service	Community Residential Rehab facility	Male = 56% White = 97.5%	1358	Abstinence, addiction severity and MH	No significantly different outcomes in abstinence rates at 1-month, older adults engaged informal post-discharge aftercare less than MA adults.
15. Outlaw et al. (2012)	CBT and self-management	Dual diagnosis service for OA public housing	Majority male and white	199	Alcohol consumption, binge drinking, stress levels	Program completers significantly decreased use of nonmedical prescription drugs, improved cognitive functioning, MH, vitality, and lack of bodily pain.
16. Poole et al. (2009)	Group therapy	Community clinic setting	Majority men and Canadian	12	Mental health and substance use.	Themes - acceptance, befriending, guilt, power, and holding on. Narrative therapy is well suited to older adults coping with mental health and substance use
17. Rao. (2014)	Targeted community nursing service	Community mental health	None mentioned	108	Alcohol consumption	108 patients aged 65 and over with alcohol misuse were identified. 50 patients were taken on by community MH teams, of whom 19 patients had achieved abstinence from alcohol or controlled drinking at the 6 months follow up 38%.
18. Schonfeld et al. (2015)	BI	Services targeting older people:	Majority females and white	85 001	Substance use	8165 clients were at moderate or high risk. Most received brief intervention for alcohol or medication misuse. Six-month follow-ups revealed a significant decrease in substance use.
19. Watson et al. (2013)	Stepped care vs. BI	Primary care	Majority male	529	Alcohol consumption	Stepped care does not confer an advantage over minimal intervention in terms of reduction in alcohol consumption Cost-effectiveness analysis suggested stepped care intervention is more likely to generate greater health benefits.

Abbreviations: BI, brief interventions; OA, older adults; MA, middle-aged adults; PSU, problematic substance use, MH, mental health.

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
