# Peer review of "Community Based Interventions for Problematic Substance Use in Later Life: A Systematic Review of Evaluated Studies and Their Outcomes"

_ijerph, 2020, doi:10.3390/ijerph17217994_

Round 1

Reviewer 1 Report

Comments:

Overall, the outcomes of this manuscript contains messages that could be useful for the targeted populations—older adults. Nonetheless, the whole manuscript is yet to be improved with more elaborations and clarifications, with consistency.

Abstract:

Research problem has been well explained, however, this section could be more specific. For example, emphasize the target group of older people, instead of just “…which aimed to meet [older] people’s needs holistically” in Line 26-27.  Besides, provide some examples of “problematic substance use” will be useful, such as the sentence in Line 45-46.

  1. Introduction:

Well written. Only some parts could be more specific, for example, in Line 75-76, …”…some people experience non-problematic use while others may develop a range of problems, such as…”

Please kindly check through the whole manuscript with the in-text citations, as some are “Error! Reference source not found” in Line 66, Line 252 and more.

  1. Materials and Methods:

Great work in most parts like search strategy, as well as inclusion and exclusion criteria with relevant information and details. It is beneficial to include both quality assessments which increase the whole quality of the systematic review.

  1. Results:

However, there are some improvements, such as in Line 270-271, please kindly revise the sentence.

Table 3.1 is very clear and gives a good overview of the papers included to be reviewed. However, please kindly revise Table 3.2, and the formatting of all tables should be aligned, with the same font type and size, as well as the overall outline.

Some other improvements are the in-text citations, please double check whether [xx], come before the period or after. As I know up to now, it should be “… [38].”, but not “….[38]”. Besides, do include the number of citation after the paper has been mentioned, e.g. in Line 298-299, “Benza et al. [number] … Outlaw et al. [number]…”.

In Line 378, the year of publication of the article by Nicholas and Roche is missing. To avoid mistakes like this, please kindly revise whole manuscript very carefully.

For content wise, explain some jargons would be useful, such as what is prevention bingo?

In Line 536, “A study compared minimal (or brief?) intervention (BI)…”.

  1. Discussions

Also, the use of abbreviations should be aligned throughout the whole manuscripts, but not sometimes “BI”, and sometimes “brief intervention” in Line 628. Same for PSU, as some in Line 647 written as words, but some as abbreviations. Please kindly revise and align this throughout whole manuscript.

There are some parts in Results should be restructured, as some of the content maybe more suitable in Discussions.

  1. Limitations

Very well explained.

  1. Conclusions

In Line 690, instead of “this population”, should be more specific in emphasizing the targeted group of this review.

In overall, I must say it is a good piece of work, however, other than the comments mentioned above, please kindly shorten long sentences to increase the comprehension and quality in general. Besides, it comes with a lot of information and thus, split this review into two different reviews with one focus on the intervention and one on the outcomes maybe an idea.

References:

This part need to revise intensively as the formatting are not aligned with missing information. If this is APA format, please revise and edit accordingly to the latest 7th edition, with all the same type font and size. Some formats, e.g. capitalize the name of the journal, e.g. Jama, should be JAMA. As well as only first word of the topic should be capitalized, but not all words e.g. number 18. Incomplete information of the citations, e.g. no 57, 58, 62 …

Author Response

Reviewer 1

Please see below a point-by-point response and updated manuscript with changes highlighted in yellow.

Comments:

Overall, the outcomes of this manuscript contains messages that could be useful for the targeted populations—older adults. Nonetheless, the whole manuscript is yet to be improved with more elaborations and clarifications, with consistency.

Thank you.  We have edited the manuscript and expanded and clarified to improve comprehension and quality.

Abstract:

Research problem has been well explained, however, this section could be more specific. For example, emphasize the target group of older people, instead of just “…which aimed to meet [older] people’s needs holistically” in Line 26-27.  Besides, provide some examples of “problematic substance use” will be useful, such as the sentence in Line 45-46.

We have reviewed this section and provided examples as suggested.

Introduction:

Well written. Only some parts could be more specific, for example, in Line 75-76, …”…some people experience non-problematic use while others may develop a range of problems, such as…”

Please kindly check through the whole manuscript with the in-text citations, as some are “Error! Reference source not found” in Line 66, Line 252 and more.

We have provided clarification as highlighted and all of the in-text citations have been reviewed and corrected.

Materials and Methods:

Great work in most parts like search strategy, as well as inclusion and exclusion criteria with relevant information and details. It is beneficial to include both quality assessments which increase the whole quality of the systematic review.

Thankyou

Results:

However, there are some improvements, such as in Line 270-271, please kindly revise the sentence.

This sentence has been revised.

Table 3.1 is very clear and gives a good overview of the papers included to be reviewed. However, please kindly revise Table 3.2, and the formatting of all tables should be aligned, with the same font type and size, as well as the overall outline.

This table has been revised and other tables checked and reformatted.

Some other improvements are the in-text citations, please double check whether [xx], come before the period or after. As I know up to now, it should be “… [38].”, but not “….[38]”. Besides, do include the number of citation after the paper has been mentioned, e.g. in Line 298-299, “Benza et al. [number] … Outlaw et al. [number]…”.

The in-text citations have all been checked and amended.

In Line 378, the year of publication of the article by Nicholas and Roche is missing. To avoid mistakes like this, please kindly revise whole manuscript very carefully.

The date has now been added to this reference

For content wise, explain some jargons would be useful, such as what is prevention bingo?

This has been explained and generally we have clarified further throughout the manuscript.

In Line 536, “A study compared minimal (or brief?) intervention (BI)…”.

This sentence has been amended.

Discussions

Also, the use of abbreviations should be aligned throughout the whole manuscripts, but not sometimes “BI”, and sometimes “brief intervention” in Line 628. Same for PSU, as some in Line 647 written as words, but some as abbreviations. Please kindly revise and align this throughout whole manuscript.

We have avoided the use of abbreviations in our revisions, giving full terms where appropriate.

There are some parts in Results should be restructured, as some of the content maybe more suitable in Discussions.

Thank you, we have revised where relevant.

Limitations

Very well explained.

Conclusions

In Line 690, instead of “this population”, should be more specific in emphasizing the targeted group of this review.

This sentence has been revised.

In overall, I must say it is a good piece of work, however, other than the comments mentioned above, please kindly shorten long sentences to increase the comprehension and quality in general. Besides, it comes with a lot of information and thus, split this review into two different reviews with one focus on the intervention and one on the outcomes maybe an idea.

Thankyou, we have reviewed the manuscript.  We considered this but as the aim of the paper is to identify interventions that have been formally evaluated, it was not feasible to separate this into two papers as the aim is closely related.

References:

This part need to revise intensively as the formatting are not aligned with missing information. If this is APA format, please revise and edit accordingly to the latest 7th edition, with all the same type font and size. Some formats, e.g. capitalize the name of the journal, e.g. Jama, should be JAMA. As well as only first word of the topic should be capitalized, but not all words e.g. number 18. Incomplete information of the citations, e.g. no 57, 58, 62 …

These have been revised.

Reviewer 2 Report

Thank you for allowing me to read your paper.  I thoroughly enjoyed reading it and have few comments to make on it, mostly technical/process matters

The abstract is very clear and summarises well the content of the paper.

Introduction

The introduction is well written and provides a comprehensive background to the review questions.  

One very small point is that some strange text appears twice in the paper, first at p2 line 66 and relates to an error message around referencing.  (p9, line 52 is the other occurrence of this that I noticed)

The last comment that I have to make on this section is that the rationale for excluding hospital based treatment could be made a bit clearer - just even adding one sentence.

Methods and Materials

Again this section in my view was well structured and explained, and presented tables in a clear way that matches the study's RQs. 

The only issue on this section in my view relates to the PRISMA diagram where the numbers dont add up. Specifically, when the authors report the figures regarding exclusion from the 189 included studies (after COVIDENCE automatically excluded 57 papers).  After this the diagram states 73 records were then excluded which should bring the included papers down to 116 (189-73).  So there's something not quite right there.  In addition in the box which lists the 73 excluded papers by type - if we add up the numbers in that box then the total is 70.  So, again there's an issue with how the final total of 19 included papers was arrived at.  Last even with what is reported then 93 papers minus 73 stated exclusions leaves 20 rather than 19.  This all needs clarification by the authors. 

Discussion and limitations

The discussion read well, was apt and covered the pertinent points in my view.  One very small point is that while I understand perfectly what is meant by baby boomers, this might not be understood by an international audience.

Author Response

Reviewer 2

Thank you for allowing me to read your paper.  I thoroughly enjoyed reading it and have few comments to make on it, mostly technical/process matters

The abstract is very clear and summarises well the content of the paper.

Introduction

The introduction is well written and provides a comprehensive background to the review questions.  

One very small point is that some strange text appears twice in the paper, first at p2 line 66 and relates to an error message around referencing.  (p9, line 52 is the other occurrence of this that I noticed)

This has been corrected.

The last comment that I have to make on this section is that the rationale for excluding hospital based treatment could be made a bit clearer - just even adding one sentence.

We have added a sentence to clarify, thanks.

Methods and Materials

Again this section in my view was well structured and explained, and presented tables in a clear way that matches the study's RQs. 

The only issue on this section in my view relates to the PRISMA diagram where the numbers dont add up. Specifically, when the authors report the figures regarding exclusion from the 189 included studies (after COVIDENCE automatically excluded 57 papers).  After this the diagram states 73 records were then excluded which should bring the included papers down to 116 (189-73).  So there's something not quite right there.  In addition in the box which lists the 73 excluded papers by type - if we add up the numbers in that box then the total is 70.  So, again there's an issue with how the final total of 19 included papers was arrived at.  Last even with what is reported then 93 papers minus 73 stated exclusions leaves 20 rather than 19.  This all needs clarification by the authors. 

The PRISMA has been amended, thank you for drawing attention to this.

Discussion and limitations

The discussion read well, was apt and covered the pertinent points in my view.  One very small point is that while I understand perfectly what is meant by baby boomers, this might not be understood by an international audience.

We have now amended a sentence to explain this.

Reviewer 3 Report

The manuscript is interesting but several Key Elements deserve more attention.

1) the authors have opted not to include gray literature. while this has some merit, it perhaps could be considered a limitation as well. Please elaborate

2) Introduction basically needs to be rewritten and should be lenghtier

3) the proposed search strategy is neither comprehensive nor sophisticated. We suggest the authors to consult a health information specialist to improve its design.

Please consider the Peer Review of Electronic Search Strategies (PRESS)

Author Response

Reviewer 3

Thank you for your comments. We have now updated the manuscript and highlighted updated sections yellow.

The manuscript is interesting but several Key Elements deserve more attention.

The authors have opted not to include gray literature. while this has some merit, it perhaps could be considered a limitation as well. Please elaborate

We have now explained this further for clarification in the introduction and methodology, ‘study selection’ section.

2) Introduction basically needs to be rewritten and should be lengthier

This is already quite a length paper and we have adhered to the author guidelines given below which ask for a brief introduction of reviews.  Without specific advice about what the reviewer thinks is missing, we are unable to expand.

“Introduction – should briefly place the study in a broad context and highlight why it is important.  It should define the purpose of the work and its significance including specific hypotheses being tested.  The current state of the research field should be reviewed carefully and key publications cited.    Keep the introduction comprehensible to scientists working outside the topic of the paper?”

3) the proposed search strategy is neither comprehensive nor sophisticated. We suggest the authors to consult a health information specialist to improve its design.

Please consider the Peer Review of Electronic Search Strategies (PRESS)

We are not sure what to do in relation to this feedback as this contradicts the other 2 reviewers feedback.  Secondly, there is no specific information as to why it is ‘not sophisticated’ enough.  We reiterate that our review protocol was peer reviewed and registered with Prospero.  We are not able to redesign the search at this stage.